# Rhabdomyolysis after COVID-19 Infection: A Case Report and Review of the Literature

**DOI:** 10.3390/v14102255

**Published:** 2022-10-14

**Authors:** Monica Bawor, Shwetha Sairam, Rachel Rozewicz, Stuart Viegas, Alexander N. Comninos, Ali Abbara

**Affiliations:** 1Department of Medicine, Imperial College Healthcare NHS Trust, London W6 8RF, UK; 2Division of Neurology, Imperial College London, London SW7 2BX, UK; 3Division of Diabetes, Endocrinology, and Metabolism, Imperial College London, London SW7 2BX, UK; 4Department of Endocrinology, Imperial College Healthcare NHS Trust, London W6 8RF, UK

**Keywords:** rhabdomyolysis, viral-induced rhabdomyolysis, COVID-19 disease, SARS-CoV-2 infection

## Abstract

Rhabdomyolysis is a condition in which muscle breaks down potentially leading to renal dysfunction, and often occurs secondary to a precipitating factor. Viral or bacterial infections are common precipitants for initiating rhabdomyolysis. Recently, healthcare systems across the world have been challenged by a pandemic of Severe Acute Respiratory Syndrome Coronavirus 2 (SARS-CoV-2) causing ‘coronavirus disease 2019’ (COVID-19) disease. SARS-CoV-2 infection is recognized to cause respiratory and cardiovascular compromise, thromboembolic events, and acute kidney injury (AKI); however, it is not known whether it can precipitate rhabdomyolysis, with only a limited number of cases of SARS-CoV-2 infection preceding rhabdomyolysis reported to date. Here, we report the case of a 64-year-old woman who developed rhabdomyolysis shortly after SARS-CoV-2 infection and COVID-19. She initially presented with muscular pain, a creatine kinase level of 119,301 IU/L, and a mild rise in her creatinine level to 92 µmol/L, but successfully recovered with intravenous fluid support. We also review the literature to summarise previously reported cases of rhabdomyolysis precipitated by SARS-CoV-2, highlighting the need to consider this diagnosis in patients presenting with SARS-CoV-2 and myalgia.

## 1. Introduction

Since its first emergence in 2019, SARS-CoV-2 infection causing COVID-19 disease has been associated with numerous complications including respiratory and cardiovascular compromise, thromboembolic events, and acute kidney injury (AKI) [1,2]. The incidence of rhabdomyolysis after COVID-19 is not known, and this association has only been described in a few case reports to date. Viral infections are a known precipitant of rhabdomyolysis [3], and myalgia is a common presenting feature of both COVID-19 and rhabdomyolysis. Here, we present a case of rhabdomyolysis in a patient with COVID-19 illness highlighting the need to consider this diagnosis in patients presenting with both COVID-19 and myalgia.

## 2. Case Presentation

A 64-year-old Afro-Caribbean woman with a background history of hypertension and previous infrequent episodes of viral-induced rhabdomyolysis presented to the emergency department with bilateral thigh pain and dark urine for one day. She reported a 5-day history of cough, lethargy, reduced appetite, and taste disturbance. She had tested positive for COVID-19 infection on lateral flow test (variant not specified) on the day of her admission. She had declined vaccination for SARS-CoV-2. 

On further assessment, the patient denied nausea, vomiting, or any gastrointestinal, urinary, or neurological symptoms. She did not report any fevers and was apyrexial on admission. On physical examination, the patient has a blood pressure of 139/88 mmHg, a pulse rate of 88 beats per minute, and a respiratory rate of 16 breaths per minute. Her pulse oximetry reading reported an oxygen saturation of 97% on room air. She was alert and had normal vesicular breathing on pulmonary examination and the remainder of her systems examinations were unremarkable.

## 3. Background History and Previous Presentations of Rhabdomyolysis

Our patient had a history of infrequent episodes of rhabdomyolysis in the past on five occasions, all of which were triggered by viral infections. The first was in 1987 which was triggered by a suspected viral upper respiratory tract infection (URTI), although the specific organism was not identified. Following this she was noted to have had a normal muscle biopsy. The second episode was in 2000 also preceded by a suspected viral URTI. The third episode in 2013 was attributed to a presumed viral URTI as well as dental infection. The fourth episode was in 2018 triggered by the Influenza virus causing an URTI; she had mild muscle ache however she did not seek medical attention. The fifth episode in 2019 was also triggered by the Influenza virus causing an URTI. This patient had two additional viral infections (2015 and 2016) but experienced no muscle pain or neurological sequelae and did not seek medical attention. 

During the episode of rhabdomyolysis in 2016, her creatine kinase (CK) level was 205 IU/L (Reference Range; RR 25–200 IU/L), erythrocyte sedimentation rate (ESR) was 51 mm/h (RR 5.0–15 mm/h) and C-reactive protein (CRP) was 10 mg/L (RR 0–5 mg/L). Her antinuclear antibody (ANA) was borderline positive (1:160) with a homogenous pattern. antineutrophil cytoplasmic antibody (ANCA), extractable nuclear antigen (ENA) and myositis immunoblot were all negative. In 2017, DNA samples analysed in Sheffield for extended rhabdomyolysis gene panel testing as well as molecular analysis, did not reveal any significant genetic predisposition variants. In 2019, her CK levels reached a peak at 36,320 IU/L during an episode of mild rhabdomyolysis triggered by a viral infection. Further tests including a non-ischemic forearm test, Pompe disease tests, and urinary organic acid, were all within normal limits.

## 4. Investigations

During the admission, blood test results showed an initial creatine kinase level of 119,301 IU/L (RR 25–200 IU/L). Her creatinine level was 92 µmol/L (RR 45–84 µmol/L) with an estimated glomerular filtration rate (eGFR) of 82 mL/min/1.73 m^2^ (RR > 90 mL/min/1.73 m^2^). Her platelet count was mildly reduced at 114 × 10^9^/L (RR 150–450 × 10^9^/L). Her serum ferritin level was within the reference range at 246 µg/L (RR 24–307 µg/L), serum vitamin B12 level was mildly elevated at 804 ng/L (RR 200–800 ng/L) and folate was elevated at 14 µg/L (RR 1.8–9 µg/L). Pertinent blood test results following the day of admission (Day 0) are presented in Table 1 and Figure 1 below.

Creatine kinase levels were highest on admission at 119,301 IU/L and reduced significantly within five days to 20,000 IU/L. At two weeks follow-up her CK levels had returned to within the reference range. With regard to her renal function, her creatinine was mildly elevated on admission but did not deteriorate following treatment with intravenous fluid support. 

We undertook a full renal screen: Immunoglobulin testing revealed levels of IgA 1.22 g/L (RR 0.80–4.00 g/L), IgG 10.9 g/L (RR 6.0–16 g/L), IgM 0.27 g/L (RR 0.50–2.00 g/L). Serum free light chains were within normal limits; kappa light chains were 17.8 mg/L (RR 3.3–19.4 mg/L), lambda light chains were 15 mg/L (RR 5.71–26.3 mg/L), and kappa to lambda ratio was 1.19 (RR 0.26–1.65). Protein electrophoresis was normal. Rheumatoid factor was undetectable (RR < 0.20 IU/L). Autoantibody serology testing was negative for ANA, ANCA, serine proteinase 3 antibody (PR3), and myeloperoxidase (MPO). Viral serology indicating active infection with cytomegalovirus (CMV), Epstein–Barr virus (EBV), human immunodeficiency virus (HIV), Hepatitis A, Hepatitis B, Hepatitis C, and parvovirus were all not detected. 

Urine dip showed blood (3+), protein (3+), leukocytes negative, nitrites negative, ketones negative and pH 5. Chest X-ray on admission showed clear lungs and pleural spaces (Figure 2). 

The patient did not exhibit any radiographic evidence of COVID-19 disease and had presented with mild respiratory symptoms only, mainly cough. 

## 5. Treatment

The patient was admitted to a medical ward where she received treatment with continuous intravenous (IV) fluids with 4 L per day. She was also treated with 40 mg of enoxaparin subcutaneously for prophylaxis of venous thrombosis. During her admission she also received calcium-vitamin D (Adcal D3) tablets for hypocalcaemia, potassium bicarbonate-potassium chloride (Sando-K) tablets for hypokalaemia, and sodium dihydrogen phosphate (Sando-Phos) tablets for hypophosphatemia. In addition, she received paracetamol for analgesia, and a phosphate enema for constipation. Her regular medications, including losartan and indapamide, were held during admission to prevent kidney dysfunction.

## 6. Outcome and Follow-Up

The patient’s clinical condition remained stable, and her levels of creatine kinase decreased on daily monitoring without any suggestion of abnormal renal function. As her CK continued to fall to 33,000 IU/L by the fifth day of her admission, she was discharged and asked to continue good hydration with oral fluid intake. She had follow-up at 9 days following discharge to confirm that her creatine kinase levels continued to decrease (CK 1709 IU/L). She was also followed up in the neuromuscular clinic 1 month following admission, when was noted to have some mild residual proximal muscle weakness in her left leg. 

## 7. Discussion

Rhabdomyolysis is a known complication of viral and bacterial infections [3]. It is characterized by the breakdown of skeletal muscle leading to the release of muscular components into the blood; including myoglobin, creatine kinase and lactate dehydrogenase (LDH) [3]. Rhabdomyolysis presents on a spectrum of illness severity ranging from being asymptomatic to life-threatening, and can be associated with electrolyte derangement, impairment of renal function, and rarely disseminated intravascular coagulation (DIC) [4]. It is commonly precipitated by direct injury or trauma; however, it can also be caused by drugs, toxins, infections, metabolic or genetic conditions, muscle ischemia, or hyperthermic states [4]. It is thought that viral-induced rhabdomyolysis occurs due to viral invasion of muscle tissue, toxin-induced damage, innate inflammatory responses, or all of these mechanisms in combination [3]. Patients often present with muscle pain or weakness, swelling, tea-coloured urine, and have significantly elevated levels of creatine kinase [5].

### 7.1. Risk Factors for Rhabdomyolysis in COVID-19 Illness 

Rhabdomyolysis has been associated with COVID-19 infection in previous reports [6,7,8,9,10,11,12,13,14,15,16,17,18,19,20,21,22], which we have summarised in Table 2. Although there is limited information with regard to the association between SARS-CoV-2 infection and rhabdomyolysis, there seems to be no direct correlation between the severity of COVID-19 illness and the incidence of rhabdomyolysis. Studies from early in the pandemic suggested an association between rhabdomyolysis after COVID-19 illness and an increased risk of morbidity, ICU admission, and in-hospital mortality [23]. However, this could simply reflect rhabdomyolysis being more common in patients who are critically unwell with longer hospital stays [24,25]. A study including 140 patients with rhabdomyolysis and COVID-19 found that there was an increased risk of requiring renal replacement therapy and increased risk of death [26]; however, the prognosis is favourable amongst patients with normal renal function such as in our case.

In our summary of reported cases, 34 of 38 (87%) were male, with an average age of 55.9 years, 7 of the 17 cases with known ethnicity were Black, and presented with an average CK level of 70,250 IU/L. Notably, 17 of the 38 cases (44.7%) died and 2 of the remaining survivors required haemodialysis on discharge. Thus, it is possible that rhabdomyolysis following SARS-CoV-2 could occur more commonly in patients with an increased risk of death. 

It is unclear why some patients develop rhabdomyolysis whereas others do not, and risk factors for rhabdomyolysis in this context have not been clearly defined. Rhabdomyolysis after SARS-CoV-2 infection has been most commonly documented among older males [21]. However, there does not appear to be any correlation with ethnicity or medical background in our summary of published case reports (Table 2). 

### 7.2. Viral-Induced Rhabdomyolysis 

Our patient has had previous episodes of rhabdomyolysis in the past, precipitated by viral infection. However, these have been infrequent with four episodes over four decades. Nevertheless, she may well be predisposed to rhabdomyolysis in response to viral infection. Nonetheless, despite thorough investigation, to date, no predisposing factor has been identified in this patient. 

This phenomenon of recurrent rhabdomyolysis has been observed in another case of a young Afro-Caribbean male [22] with significantly raised CK (>100,000 IU/L) who was also investigated for a metabolic myopathy; however, they too were unable identify the aetiology of recurrent viral-induced rhabdomyolysis. 

### 7.3. Rhabdomyolysis and COVID-19 Vaccination

Data regarding vaccine-induced rhabdomyolysis is limited however some studies have described cases of rhabdomyolysis related to mRNA COVID-19 vaccination including Pfizer-BioNTech and Moderna vaccines [27,28,29,30,31]. These patients presented with muscle pain and fatigue up to two weeks after vaccination. One particular study identified a patient who developed rhabdomyolysis following COVID-19 infection due to a ryanodine receptor 1 (RYR1) gene mutation, which is known to increase susceptibility to rhabdomyolysis [32]. 

A study investigated the association between COVID-19 vaccination and rhabdomyolysis using the Vaccine Adverse Event Reporting System (VAERS) and found that rhabdomyolysis following COVID-19 vaccination was more frequently reported compared to rhabdomyolysis following all other vaccinations; however, the rates remained within the expected incidence range for the general population [33]. They concluded that there is no significant increase in risk of rhabdomyolysis following COVID-19 vaccination. Furthermore, rates of rhabdomyolysis were not found to differ between different vaccines including Pfizer, Moderna, or Johnson & Johnson. Vaccine-induced rhabdomyolysis has been observed with other viral infections including H1N1 and influenza [34,35,36]. It was postulated that patients on statin therapy for hyperlipidaemia are predisposed to rhabdomyolysis and that the influenza vaccine had triggered its onset, which may also be the case for COVID-19 vaccine-related rhabdomyolysis. Nevertheless, our patient did not receive the SARS-CoV-2 vaccination and as such, this was not a factor.

### 7.4. “Long COVID” and Musculoskeletal Involvement 

The musculoskeletal system is commonly affected by COVID-19 infection and causes symptoms including myalgia, muscle weakness, and arthralgia. Such symptoms tend to disappear as the patients recover from COVID-19 illness however they have been found to persist past the acute phase of the infection [37,38]. “Long COVID” is the term used to describe the persistence of COVID-19 symptoms for longer than four weeks after the acute phase has subsided and has been documented among patients who recovered from both mild and severe forms of COVID-19 infection. It is thought that “Long-COVID” is caused by a persistent pro-inflammatory state and can therefore exacerbate muscle pain leading to prolonged intolerance of physical activity [39]. 

In one report, a young female suffering from “Long-COVID” with persistent muscle weakness and exercise intolerance was investigated with muscle biopsy, which was found to be suggestive of critical illness myopathy (CIM) despite having a relatively mild acute COVID-19 infection [40]. However, there are no documented reports of an association between “Long-COVID” and rhabdomyolysis. 

## 8. Conclusions

We report the case of a patient with rhabdomyolysis as a complication of COVID-19 illness associated with myalgia, fatigue, and significantly elevated CK levels. This 64-year-old woman had SARS-CoV-2 infection with mild symptoms and successful recovery with intravenous fluid support with no impairment of renal function. There have been reports to suggest that vaccination against SARS-CoV-2 could also precipitate rhabdomyolysis in some patients; however, this has not been proven to be a common adverse effect of vaccination and equally it is also possible that vaccination could reduce complication rates after SARS-CoV-2 infection. Finally, COVID-19 illness can lead to long-term muscle pain and weakness; however, it has not been reported to be specifically associated with rhabdomyolysis. Thus, our case report suggests that rhabdomyolysis should be considered in patients presenting with SARS-CoV-2 infection, especially in the presence of myalgia, or if there is urinary discoloration.

## Figures and Tables

**Figure 1 viruses-14-02255-f001:**
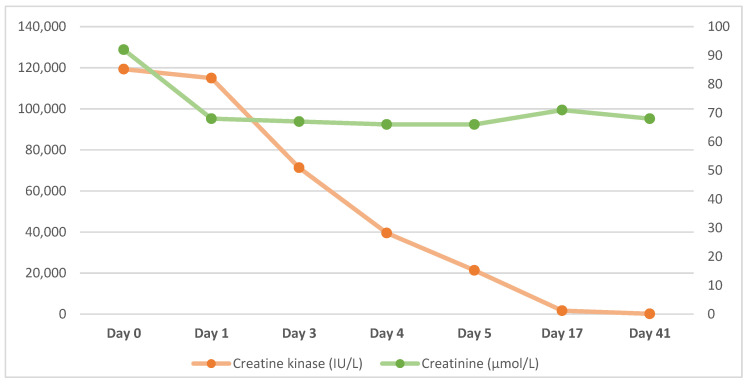
Creatine kinase (left axis) and creatinine levels (right axis) during admission and at follow up.

**Figure 2 viruses-14-02255-f002:**
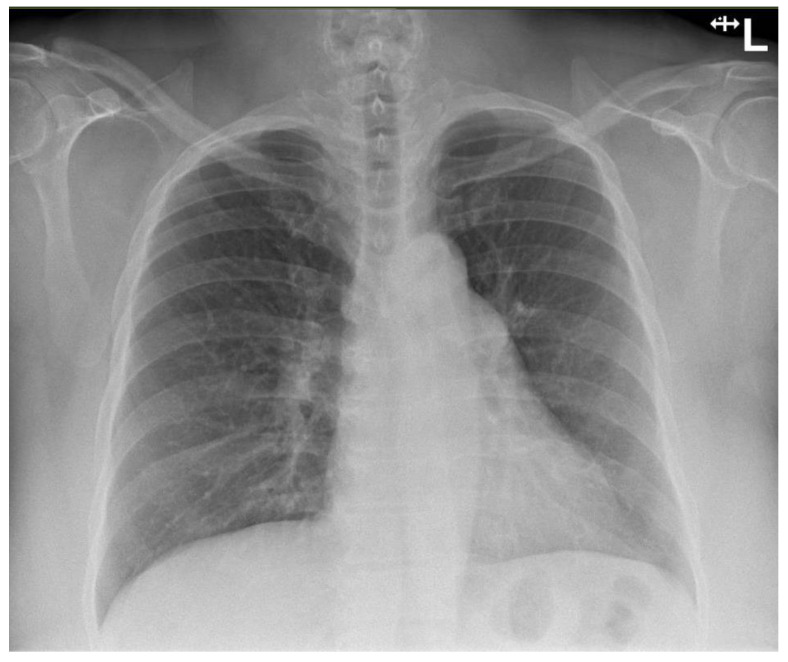
Normal chest X-ray radiograph on admission.

**Table 1 viruses-14-02255-t001:** Blood test results during admission and at follow up. Day 0 indicates day of presentation and admission to hospital.

Date	Hb (g/L)	WCC (×10^9^/L)	Creatinine (µmol/L)	eGFR (ml/min/1.73 m^2^)	CRP (mg/L)	Platelets (×10^9^/L)	Creatine Kinase (IU/L)
Day 0	137	4.6	92	57	-	114	119,301
Day 1	123	3.2	68	82	11.9	72	114,965
Day 2	125	3.5	67	83	8.4	68	71,303
Day 3	128	4.2	66	85	10.3	79	39,540
Day 4	116	3.8	66	85	8.7	106	21,344
Day 10	126	5.4	71	78	-	314	1709
Day 41	-	-	68	82	-	-	195

**Table 2 viruses-14-02255-t002:** Summary of case reports in the literature describing rhabdomyolysis in COVID-19.

	Age (Years)	Gender	Ethnicity	Background	Level of Care	Peak CK Level (IU/L)	AKI	Outcome
Valente-Acosta 2020 [6]	71	Male	Unknown	BPH, smoker	Ward	8720	Yes	Discharged
Taxbro 2020 [7]	38	Male	Unknown	T2DM, gout, mild obesity	ICU	N/A	Yes	Discharged
Fujita 2021 [8]	19	Female	Unknown	None	Ward	55,613	No	Discharged
Jin & Tong 2020 [9]	60	Male	Unknown	Unknown	Ward	11,842	No	Discharged
Suwanwongse & Shabarek 2020 [10]	88	Male	Unknown	HTN, CKD, HF, BPH, OA, Cognitive impairment	Ward	13,581	Yes	Discharged
Uysal 2020 [11]	60	Male	Unknown	None	Ward	4267	No	Discharged
Singh 2020 [12]—Case 1	67	Male	Unknown	HTN	ICU	19,773	Yes	Died
Singh 2020 [12]—Case 2	39	Male	Unknown	HTN	ICU	4330	Yes	Died
Singh 2020 [12]—Case 3	43	Male	Unknown	ESRF	Ward	9793	Unknown	Died
Singh 2020 [12]—Case 4	70	Male	Unknown	None	ICU	5008	Unknown	Died
Khosla 2020 [13]—Case 1	65	Male	Black	HTN, OSA hyperlipidaemia,	ICU	7854	Yes	Died
Khosla 2020 [13]—Case 2	78	Male	White	T2DM, HTN, hyperlipidaemia, HF, MVR, CABG	ICU	>22,000	Yes	Died
Khosla 2020 [13]—Case 3	67	Male	Black	T2DM, HTN, hyperlipidaemia CKD, hemicolectomy	Dialysis	6164	Yes	Discharged
Khosla 2020 [13]—Case 4	58	Male	Black	T2DM, HTN, hyperlipidaemia	Ward	4625	Yes	Discharged
Khosla 2020 [13]—Case 5	64	Male	Black	T2DM, HTN, HIV	Ward	3135	No	Discharged
Rivas-Garcia 2020 [14]	78	Male	White	HTN, T2DM	Ward	22,511	Yes	Discharged
Hussein 2020 [15]	38	Male	Unknown	Obesity	ICU	33,000	Yes	Discharged
Alrubaye & Choudhury 2020 [16]	35	Female	Unknown	None	Ward	71,000	Unknown	Discharged
Buckholz 2020 [17]—Case 1	43	Male	Unknown	None	ICU	75,240	Yes	Discharged
Buckholz 2020 [17]—Case 2	37	Male	Unknown	None	ICU	82,960	Yes	Discharged
Buckholz 2020 [17]—Case 3	75	Male	Unknown	DVT	Ward	3638	Yes	Discharged
Buckholz 2020 [17]—Case 4	59	Male	Unknown	None	ICU	8310	No	Died
Buckholz 2020 [17]—Case 5	66	Male	Unknown	HTN	ICU	10,100	No	Discharged
Buckholz 2020 [17]—Case 6	70	Female	Unknown	MM, CKD	ICU	406,300	Yes	Died
Solis 2020 [18]	46	Male	Unknown	CML	Unknown	400,000	Yes	Died
Chedid 2020 [19]	51	Male	Unknown	HTN, T2DM, CKD	Unknown	464,000	Yes	Discharged on HD
Byler 2021 [20]	67	Female	Unknown	HTN, T2DM, Hyperlipidaemia	ICU	15,085	Yes	Discharged on HD
Singh 2020 [21]—Case 1	54	Male	Hispanic	Asthma, DM, HTN, obesity	Unknown	7337	Unknown	Died
Singh 2020 [21]—Case 2	54	Male	White	None	Unknown	3068	Unknown	Discharged
Singh 2020 [21]—Case 3	34	Male	White	Obesity, prediabetes	Unknown	5454	Unknown	Died
Singh 2020 [21]—Case 4	71	Male	Black	HTN, seizures schizophrenia,	Unknown	10,247	Unknown	Died
Singh 2020 [21]—Case 5	88	Male	White	Diabetes, HTN	Unknown	2628	Unknown	Died
Singh 2020 [21]—Case 6	56	Male	Hispanic	HTN, prediabetes	Unknown	5388	Unknown	Discharged
Singh 2020 [21]—Case 7	57	Male	Hispanic	None	Unknown	37,524	Unknown	Died
Singh 2020 [21]—Case 8	64	Male	White	None	Unknown	6435	Unknown	Died
Singh 2020 [21]—Case 9	36	Male	White	None	Unknown	5531	Unknown	Died
Singh 2020 [21]—Case 10	39	Male	Black	HTN	Unknown	4330	Unknown	Died
Shanbhag 2020 [22]	19	Male	Black	Anxiety, previous influenza-associated rhabdomyolysis	Unknown	694,200	No	Discharged

*AC = Afro-Caribbean, BPH = benign prostatic hyperplasia; T2DM = Type 2 diabetes mellitus; ICU = intensive care unit; HTN = hypertension; CKD = chronic kidney disease; HF = heart failure; OA = osteoarthritis; ESRF = end-stage renal failure; OSA = obstructive sleep apnoea; MVR = mitral valve repair; CABG = coronary artery bypass graft; HIV = Human Immunodeficiency Virus; DVT = deep vein thrombosis; MM = multiple myeloma; CML = chronic myeloid leukaemia; Haemodialysis = HD.*

## Data Availability

Not applicable.

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
