# Peer review of "Rhabdomyolysis after COVID-19 Infection: A Case Report and Review of the Literature"

_viruses, 2022, doi:10.3390/v14102255_

Round 1
Reviewer 1 Report
Dear Editor
Thank you for your invitation as a reviewer to this article for publication in Viruses. I congratulate the authors for addressing this important and current issue, but there are points that need improvement.
My suggestions for the article are as follows:
1. "Table 2" is mentioned 2 times in the text, but there is only 1 table. Table 2 should be added.
2. There is "Figure 2" but it is not mentioned anywhere in the text, it should be mentioned.
3. References in the manuscript begin in section "7.1" of the discussion. However, there is general information that needs to be referenced in the first paragraphs of the introduction and discussion. References on these general topics should be added.‘
Best Regards
Reviewer 2 Report
I congratulate the authors for compiling all the articles on Covid-19 illness-Rhabdomyolysis and presenting their interesting cases very well.
Minor revision
In 7.1. "Risk factors for rhabdomyolysis in COVID-19 illness"
Table 2 is not included in the article. Table 2 must be added to the article.
Reviewer 3 Report
Bawor et. al. presents a case of a woman who had rhabdomyolysis and COVID-19 mild infection. The case is well-written and interesting. Also, the literature review is comprehensive. I want to suggest to the authors that they should specify which kind of previous viral infectious the woman has had before. If they are not able to describe specifically the kind of virus at least the syndrome involved (for example upper respiratory syndrome) and then state in the manuscript the phrase "presumed viral infections".
Round 2
Reviewer 1 Report
The arrangements I requested were made by the authors.